# Circumventing the Uncertainties of the Liquid Phase in the Compositional Control of VLS III–V Ternary Nanowires Based on Group V Intermix

**DOI:** 10.3390/nano14020207

**Published:** 2024-01-17

**Authors:** Vladimir G. Dubrovskii

**Affiliations:** Faculty of Physics, St. Petersburg State University, Universitetskaya Emb. 13B, 199034 St. Petersburg, Russia; dubrovskii@mail.ioffe.ru

**Keywords:** III–V ternary nanowires, VLS growth, composition, vapor–solid distribution, modeling

## Abstract

Control over the composition of III–V ternary nanowires grown by the vapor–liquid–solid (VLS) method is essential for bandgap engineering in such nanomaterials and for the fabrication of functional nanowire heterostructures for a variety of applications. From the fundamental viewpoint, III–V ternary nanowires based on group V intermix (InSb_x_As_1−x_, InP_x_As_1−x_, GaP_x_As_1−x_ and many others) present the most difficult case, because the concentrations of highly volatile group V atoms in a catalyst droplet are beyond the detection limit of any characterization technique and therefore principally unknown. Here, we present a model for the vapor–solid distribution of such nanowires, which fully circumvents the uncertainties that remained in the theory so far, and we link the nanowire composition to the well-controlled parameters of vapor. The unknown concentrations of group V atoms in the droplet do not enter the distribution, despite the fact that a growing solid is surrounded by the liquid phase. The model fits satisfactorily the available data on the vapor–solid distributions of VLS InSb_x_As_1−x_, InP_x_As_1−x_ and GaP_x_As_1−x_ nanowires grown using different catalysts. Even more importantly, it provides a basis for the compositional control of III–V ternary nanowires based on group V intermix, and it can be extended over other material systems where two highly volatile elements enter a ternary solid alloy through a liquid phase.

## 1. Introduction

Control over the composition of III–V ternary materials and III–V heterostructures is required for bandgap engineering and has been a subject of extensive research for decades [1,2]. More recently, III–V ternary nanowires (NWs) and NW-based heterostructures have attracted great interest due their fundamental properties and potential applications in Si-integrated optoelectronics, quantum communication technologies and other fields [2,3,4,5,6]. Most III–V NWs are grown using different epitaxy techniques via the VLS method using a catalyst droplet, often Au [7], which can be replaced with a group III metal (Ga) in the self-catalyzed VLS approach [8]. The VLS growth of a ternary A_x_B_1−x_C NW is a complex process whereby the vapor phase containing A, B and C species condenses in a quaternary liquid phase consisting of A, B, C and Au atoms (in the case of a Au catalyst) and then crystallizes into a ternary A_x_B_1−x_C NW [9,10,11,12,13,14,15,16,17]. Due to the presence of a catalyst droplet, whose composition is generally unknown, the compositional control of VLS III–V ternary NWs remains a challenging task [10,11,12,13,14,15,16,17]. Full understanding of the VLS growth of III–V ternary NWs, particularly those based on group V intermix, has not been achieved hitherto. In this work, we try to develop a model which fully circumvents the uncertainties of the liquid phase, and we link the stationary composition of VLS III–V ternary NWs based on group V intermix to the well-controlled parameters of vapor.

The key parameters and factors influencing the composition of III–V ternary NWs grown by the VLS method are introduced as follows [9,10,11,12,13,14,15,16,17]. The composition of a quaternary liquid in a catalyst droplet is described by three independent variables, for example, (i) the fraction of A atoms in liquid,
(1)$$y=\frac{\chi_{A}}{\chi_{A}+\chi_{B}},$$
where $\chi_{A}$ and $\chi_{B}$ are the atomic concentrations of A and B atoms in liquid; (ii) the total concentration of A and B atoms in liquid, $\chi_{tot}=\chi_{A}+\chi_{B}$; and (iii) the concentration of C atoms in liquid $\chi_{C}$, with $\chi_{Au}=1-\chi_{C}-\chi_{tot}$. In the self-catalyzed VLS growth, the droplet is a ternary alloy, and the number of independent variables is reduced to two, in view of $\chi_{Au}=0$. The vapor phase, producing three atomic fluxes of A, B and C atoms $I_{A}$, $I_{B}$ and $I_{C}$, can be described by the fraction of A atoms in vapor,
(2)$$z=\frac{I_{A}}{I_{A}+I_{B}},$$
where the total flux of A and B atoms $I_{tot}=I_{A}+I_{B}$, and the flux ratio is $\left(I_{A}+I_{B}\right)/I_{C}$. The liquid–solid distribution $x\left(y\right)$ links the solid and liquid composition, whereas the vapor–solid distribution $x\left(z\right)$ links the solid and vapor composition [10,12].

Most models for the composition of VLS III–V ternary NWs developed so far treat the liquid–solid growth and hence the liquid–solid distributions, considering liquid as an isolated mother phase without any material exchange with vapor [11,12,13,14,15,16,17]. Here, we study VLS ternary NWs based on group V intermix, with the A and B atoms belonging to group V and the C atoms belonging to group III. In this case, the liquid–solid growth occurs under group-III-rich conditions, because the total concentration of highly volatile group V atoms in the droplet, $\chi_{A}+\chi_{B}\;$, is always much smaller than $\chi_{C}$. According to Ref. [10], this yields the kinetic liquid–solid distribution of III–V ternary NWs based on group V intermix, given by
(3)$$\begin{array}{l} y=\frac{x+g\left(x\right)}{c_{l}+\left(1-c_{l}\right)x},\;g\left(x\right)=\left(1-x\right)x\Gamma_{l}\left[c_{l}e^{\omega {\left(1-x\right)}^{2}}-\beta_{l}e^{\omega x^{2}}\right],\\ c_{l}=\frac{D_{A}}{D_{B}}e^{\psi_{A}-\psi_{B}},\\ \beta_{l}=e^{\mu_{A}^{l,0}-\mu_{AC}^{s,0}-\left(\mu_{B}^{l,0}-\mu_{BC}^{s,0}\right)+\psi_{A}-\psi_{B}}=e^{\Delta \mu_{AC}^{0}-\Delta \mu_{BC}^{0}+\psi_{A}-\psi_{B}}\\ \Gamma_{l}=\frac{1}{\left(\chi_{A}+\chi_{B}\right)\chi_{C}}e^{-\left(\mu_{A}^{l,0}+\mu_{C}^{l,0}-\mu_{AC}^{s.0}\right)-\psi_{A}-\psi_{C}}=\frac{1}{\left(\chi_{A}+\chi_{B}\right)\chi_{C}}e^{-\Delta \mu_{AC}^{0}-\psi_{A}-\psi_{C}}. \end{array}$$

Here, ω is the pseudo-binary interaction parameter of AC and BC pairs in solid in thermal units; $D_{k}$ are the diffusion coefficients of k= A, B atoms in liquid; $\psi_{k}$ are the interaction terms in the chemical potentials of the A and B atoms in liquid,
(4)$$\mu_{A}^{l}=\mu_{A}^{l,0}+ln\chi_{A}+\psi_{A},\;\mu_{B}^{l}=\mu_{B}^{l,0}+ln\chi_{B}+\psi_{B};$$

$\mu_{k}^{l,0}$ are the chemical potentials of pure k= A, B and C liquids; and $\mu_{kC}^{s,0}$ are the chemical potentials of the pure solid binaries AC and BC. The expressions for the parameters $\beta_{l}$ and $\Gamma_{l}$ in Equation (3) are given in the two equivalent forms, with $\Delta \mu_{AC}^{0}=\mu_{A}^{l,0}+\mu_{C}^{l,0}-\mu_{AC}^{s.0}$ and $\Delta \mu_{BC}^{0}=\mu_{B}^{l,0}+\mu_{C}^{l,0}-\mu_{BC}^{s.0}$ as the chemical potential differences for pure binaries.

The functional form of the kinetic liquid–solid distribution given by Equation (3) is the same as the kinetic vapor–solid distribution for III–V ternary materials based on group III intermix, which are grown under group-V-rich conditions without any droplet [18]. However, the coefficients in Equation (3) are modified and contain the parameters of liquid rather than vapor. While the interaction terms $\psi_{k}$ depend only on $\chi_{C}$, with neglect of small corrections containing $\chi_{A}$ and $\chi_{B}$ (see Ref. [10] for a detailed discussion), the $\Gamma_{l}$ term is inversely proportional to $\chi_{tot}=\chi_{A}+\chi_{B}$. Unfortunately, the very low concentrations of group V elements in the droplet (~0.01 or even less [9]) are below the detection limit of any characterization technique and cannot be measured during or after growth. Furthermore, there is almost no chance that the value of $\chi_{A}+\chi_{B}$ will be kept constant under varying vapor fluxes $I_{A}$ and $I_{B}$ during the VLS growth of a ternary NW, which is why even the use of $\chi_{A}+\chi_{B}$ as a fitting constant cannot be justified. This uncertainty was not circumvented in Ref. [10], where the obtained vapor–solid distribution contained $\chi_{A}+\chi_{B}$. This uncertainty makes the liquid–solid distribution given by Equation (3) almost useless for the compositional control over VLS ternary NWs based on group V intermix.

In Ref. [19], a rather general approach was developed, which resulted in the analytic vapor–solid distribution of III–V ternary materials:(5)$$\begin{array}{l} z≅\frac{x}{\varepsilon }+\left(1-\frac{1}{\varepsilon }\right)\frac{1}{1+f^{2}\left(x\right)},\\ f\left(x\right)=\beta_{g}\frac{\left(1-x\right)}{x}e^{\omega \left(2x-1\right)}. \end{array}$$

This vapor–solid distribution is the sum of the purely kinetic (z=x) and equilibrium ($z=1/[1+f^{2}\left(x\right)])$ distributions, whose weights are regulated by the effective atomic V/III ratio ε related to $\left(I_{A}+I_{B}\right)/I_{C}$. The thermodynamic function $f\left(x\right)$ contains the pseudo-binary interaction constant and the affinity parameter $\beta_{g}$, given below. When ε is close to unity, the growth of a ternary is kinetically controlled, whereas at ε≫1 the growth occurs under C-poor conditions and the vapor–solid distribution becomes close to equilibrium (or nucleation limited [14,15]). This expression fits satisfactorily the compositional data on InSb_x_As_1−x_ [2] and AlSb_x_As_1−x_ [20] epi-layers as well as Au-catalyzed VLS InSb_x_As_1−x_ NWs [21], although no droplet on the NW top was considered in the model of Ref. [19]. In Ref. [21], Borg and coauthors fitted the VLS data using Biefeld’s [2] numerical model, which is based on similar considerations as the model of Ref. [19]. Due to the additional diffusion flux of group III © atoms from the NW sidewalls to the droplet, the fitting values of the V/III ratios obtained in Refs. [19,21] are much smaller than the V/III ratios in vapor. This fundamental observation will be used in this work.

The compositions of VLS III–V ternary NWs based on group V intermix have been experimentally studied in many material systems, including InSb_x_As_1−x_ [21,22,23,24], GaSb_x_As_1−x_ [25], InP_x_As_1−x_ [26,27] and GaP_x_As_1−x_ [28,29,30,31,32], using different epitaxy techniques such as Au-catalyzed metal-organic vapor phase epitaxy (MOVPE) [21,22,25,27], Au-catalyzed chemical beam epitaxy (CBE) [26], Ag-catalyzed [26] and self-catalyzed [23,28,29,30,31] molecular beam epitaxy (MBE) on different substrates, and even the substrate-free Au-catalyzed aerotaxy by MOVPE [32] (see Refs. [11,12] for comprehensive reviews). A limited number of the measured vapor–solid distributions—for example, Au-catalyzed InP_x_As_1−x_ [26] and self-catalyzed GaP_x_As_1−x_ [29]—followed the simplest kinetic Langmuir–McLean shape (see below), with only one parameter describing the different incorporation rates of the A and B atoms into a droplet. A comprehensive experimental study by Borg and coauthors [21] revealed the transition from linear $z\left(x\right)$ dependence of Au-catalyzed InSb_x_As_1−x_ NWs at low V/III ratios to a non-linear, close-to-equilibrium shape at high V/III ratios. Such a transition was observed much earlier by Biefeld in InSb_x_As_1−x_ epi-layers [2] and predicted to be a general phenomenon in Ref. [19] (see Equation (5) above). However, the models of Refs. [2,19] considered the vapor–solid growth without any droplet, and their use for modeling the compositions of VLS NWs requires a justification.

Overall, the achieved level of the growth and compositional modeling of VLS III–V ternary NWs based on group V intermix is insufficient for quantitative comparison with the data and even for qualitative understanding of some compositional trends. The generally unknown parameters of the liquid phase should be either fully eliminated or expressed through the known parameters of vapor in the final expressions. Consequently, here we develop a fully self-consistent growth model of such NWs which, under rather general assumptions, leads to vapor–solid distributions that circumvent the uncertainties in the infinitely low group V concentrations in the droplet. It will be shown that, using some reasonable simplifications, the vapor–solid distribution can be reduced to an approximation which is very close to Equation (5), where the parameter ε accounts for the surface diffusion of group III atoms. The model fits satisfactorily the available compositional data for different VLS NWs based on group V intermix. It justifies the use of the vapor–solid distribution similar to Equation (5) for VLS NWs [21] and provides a basis for the modeling and compositional tuning of such NWs in general.

## 2. Model

We consider the steady-state VLS growth of an A_x_B_1−x_C NW based on group V intermix under the following assumptions. First, we neglect desorption of the C atoms belonging to group III from the droplet. This is usual in modeling VLS growth via MBE [9,33] and MOVPE [34] and is supported by the data of Ref. [35], showing that group III atoms can re-emit from a masked surface but not from the NW sidewalls or droplet. As a result, a NW ensemble of sufficient volume is able to collect all the group III atoms sent from vapor. The absence of group III desorption from the droplet is also supported by the measured vapor–solid distributions of III–V ternary NWs based on group III intermix, whose shape is close to the Langmuir–McLean shape in most cases [10]. Second, we assume that the droplet volume does not change over time, at least after a certain incubation stage where the measured NW composition can be different from its steady-state value. This assumption is usual in modeling of Au-catalyzed VLS growth [9,33,34]. Self-catalyzed VLS growth is different, because the droplet serves as a non-stationary reservoir of group III atoms that can either swell or shrink depending on the effective V/III ratio [36,37]. However, the droplet volume should self-equilibrate to a steady-state value, corresponding to equal group III and group V flows, and stay constant after that [36,37,38]. Third, we assume that group V atoms are not diffusive and enter NWs only through their droplets [8,36,37,38,39]. Fourth, we consider that the arriving group V species are A_2_ and B_2_ dimers, as usual in MBE [39]. This assumption is not critical. The model can be re-arranged, for example, for A_4_ and B_4_ tetramers or group V precursors containing only one group V atom, such as AsH_3_ or PH_3_. However, these precursors will most probably decompose in vapor before reaching the droplet surface, resulting in the fluxes of V_2_ dimers or V_4_ tetramers, depending on the growth temperature.

Under these assumptions, the steady-state VLS growth of a ternary NW based on group V intermix is described by the following two equations:(6)$$\begin{array}{l} x\sigma_{C}I_{C}=2\sigma_{A}I_{A_{2}}-I_{A}^{des},\\ \left(1-x\right)\sigma_{C}I_{C}=2\sigma_{B}I_{B_{2}}-I_{B}^{des}. \end{array}$$

Here, $\sigma_{A}$ and $\sigma_{B}$ are the vapor–liquid incorporation rates or, more precisely, the effective adsorption coefficients giving the ratio of the number of A or B atoms entering the droplet over the total number of these atoms impinging onto the droplet surface. They account for a possible difference in A and B beam angles in the directional deposition techniques such as MBE and include the droplet contact angle β. The $\sigma_{A}$ and $\sigma_{B}$ in our notation do not include desorption. Similarly, $\sigma_{C}$ is the effective collection efficiency of group III atoms on the droplet surface, the NW sidewalls and possibly the substrate surface. For III–V NWs, $\sigma_{C}$ may be much larger than $\sigma_{A}$ and $\sigma_{B}$, because most group III atoms are collected by the droplet from solid surfaces surrounding the droplet [9,21,26,33,34,35,36,37]. $I_{A_{2}}$ and $I_{B_{2}}$ denote the vapor fluxes of A_2_ and B_2_ dimers, bringing 2 group V atoms each, whereas $I_{A}^{des}$ and $I_{B}^{des}$ denote the desorption fluxes of the A and B atoms. The vapor composition for the fluxes of group V dimers is given by
(7)$$z=\frac{I_{A_{2}}}{I_{A_{2}}+I_{B_{2}}},$$
which is the same as Equation (2) because $I_{A}=2I_{A_{2}}$ and $I_{B}=2I_{B_{2}}$.

The equations in Equation (6) are similar to the ones considered in Ref. [10], but there is one important difference. In Ref. [10], we used the unknown NW growth rate G instead of $I_{C}$ on the left-hand side, which was then eliminated by dividing one equation by the other. This did not allow us to circumvent the uncertainty in the unknown total concentration of group V atoms in the droplet, which remained in the vapor–solid distribution. Now, the equations in Equation (6) contain the known group III flux $\sigma_{C}I_{C}$, which determines the NW growth rate in the absence of desorption. It equals the total influx of the A and B atoms minus their total desorption fluxes. This follows from summing up the two equations in Equation (6). Our aim is to express the unknown group V concentrations in the droplet $\chi_{A}$ and $\chi_{B}$ (or, equivalently, y and $\chi_{tot}=\chi_{A}+\chi_{B}$) through the vapor fluxes. To do that, we need to find the desorption fluxes as functions of $\chi_{A}$ and $\chi_{B}$. We define the desorption fluxes as the vapor fluxes which are at equilibrium with liquid at a given composition, as in Ref. [39] for a binary III–V NW. The vapor–liquid equilibrium corresponds to
(8)$$\mu_{A_{2}}^{g}=2\mu_{A}^{l},\;\mu_{B_{2}}^{g}=2\mu_{B}^{l},$$
where $\mu_{A_{2}}^{g}$ and $\mu_{B_{2}}^{g}$ are the chemical potentials of the A_2_ and B_2_ dimers in vapor.

Considering that vapor is a mixture of perfect gases, the chemical potentials of the A_2_ and B_2_ dimers are logarithmic functions of the fluxes:(9)$$\mu_{A_{2}}^{g}=2\mu_{A}^{l,0}+ln\left(\frac{I_{A_{2}}}{\sigma_{A}I_{A_{2}}^{0}}\right),\;\mu_{B_{2}}^{g}=2\mu_{B}^{l,0}+ln\left(\frac{I_{B_{2}}}{\sigma_{B}I_{B_{2}}^{0}}\right).$$

Here, we prefer to use the reference states of the A_2_ and B_2_ vapors corresponding to the fluxes $\sigma_{A}I_{A_{2}}^{0}$ and $\sigma_{B}I_{B_{2}}^{0}$ that are at equilibrium with the pure A and B liquids (having the chemical potentials $\mu_{A}^{l,0}$ and $\mu_{B}^{l,0}$). We choose the reference fluxes with the same incorporation rates $\sigma_{A}$ and $\sigma_{B}$ as for a quaternary droplet. It will be shown later that using the reference fluxes $I_{A_{2}}^{eq}=\sigma_{A}I_{A_{2}}^{0}$ and $I_{B_{2}}^{eq}=\sigma_{B}I_{B_{2}}^{0}$ (corresponding to $\sigma_{A}=\sigma_{B}=1$) does not affect the final result.

Using Equations (4) and (9) for the chemical potentials in Equation (8), we obtain the desorption fluxes in the form
(10)$$I_{A}^{des}=2I_{A_{2}}^{des}=2\sigma_{A}I_{A_{2}}^{0}e^{2\psi_{A}}\chi_{A}^{2},\;I_{B}^{des}=2I_{B_{2}}^{des}=2\sigma_{B}I_{B_{2}}^{0}e^{2\psi_{B}}\chi_{B}^{2}.$$

According to these expressions, the desorption fluxes are proportional to the squared concentrations of the A and B atoms in liquid, because group V atoms always desorb in the form of dimers [39,40]. Substitution of these desorption fluxes into Equation (6), along with the definitions for y given by Equation (1) and z by Equation (7), leads to
(11)$$\frac{I_{A_{2}}^{0}e^{2\psi_{A}}{\left(\chi_{A}+\chi_{B}\right)}^{2}y^{2}}{I_{A_{2}}+I_{B_{2}}}=z-\frac{\sigma_{C}I_{C}}{2\sigma_{A}\left(I_{A_{2}}+I_{B_{2}}\right)}x,$$
(12)$$\frac{I_{B_{2}}^{0}e^{2\psi_{B}}{\left(\chi_{A}+\chi_{B}\right)}^{2}{\left(1-y\right)}^{2}}{I_{A_{2}}+I_{B_{2}}}=1-z-\frac{\sigma_{C}I_{C}}{2\sigma_{B}\left(I_{A_{2}}+I_{B_{2}}\right)}\left(1-x\right).$$

This gives two equations for the two unknowns $\chi_{A}+\chi_{B}$ and y, which contain, however, the vapor composition z and the solid composition x. Summing up Equations (11) and (12), we find
(13)$${\left(\chi_{A}+\chi_{B}\right)}^{2}=\frac{I_{A_{2}}+I_{B_{2}}-\left(\sigma_{C}/2\sigma_{A}\right)I_{C}\left[x+c_{g}\left(1-x\right)\right]}{I_{A_{2}}^{0}e^{2\psi_{A}}y^{2}+I_{B_{2}}^{0}e^{2\psi_{B}}{\left(1-y\right)}^{2}},$$
with
(14)$$c_{g}=\frac{\sigma_{A}}{\sigma_{B}}$$
as the ratio of the vapor–liquid condensation rates of the A and B atoms. Importantly, $\chi_{A}+\chi_{B}$ is independent of the vapor composition z. However, it depends on the liquid composition y and the solid composition x, becoming x-independent only when $c_{g}=1$.

Inferring $\left(1-z\right)/z$ from Equations (11) and (12), we obtain
(15)$$z=\frac{1}{1+F},\;F=\frac{\left(\sigma_{C}/2\sigma_{B}\right)I_{C}\left(1-x\right)+I_{B_{2}}^{0}e^{2\psi_{B}}{\left(\chi_{A}+\chi_{B}\right)}^{2}{\left(1-y\right)}^{2}}{\left(\sigma_{C}/2\sigma_{A}\right)I_{C}x+I_{B_{2}}^{0}e^{2\psi_{B}}{\left(\chi_{A}+\chi_{B}\right)}^{2}{\left(1-y\right)}^{2}}.$$

Using Equation (13), after some simple manipulations, we obtain the main result of this work in the form
(16)$$z=\frac{x}{\varepsilon }+\left[1-\frac{x+c_{g}\left(1-x\right)}{\varepsilon }\right]\frac{1}{1+\zeta {\left[\left(1-y\right)/y\right]}^{2}},$$
with the parameters
(17)$$\varepsilon =\frac{2\sigma_{A}\left(I_{A_{2}}+I_{B_{2}}\right)}{\sigma_{C}I_{C}},$$
(18)$$\zeta =\frac{I_{B_{2}}^{0}}{I_{A_{2}}^{0}}e^{2\left(\psi_{B}-\psi_{A}\right)}.$$

Clearly, the parameter ε determines the effective ratio of the total flux of group V atoms over the flux of group III atoms entering the droplet. In the simplest model for surface diffusion of group III adatoms [9], the $\sigma_{A}/\sigma_{C}$ ratio is given by $\sigma_{A}/\sigma_{C}=1/(1+a\lambda_{3}/R)$, where $\lambda_{3}$ is the diffusion length of group III adatoms on the NW sidewalls, R is the NW radius and a is a constant related to the droplet contact angle β and the epitaxy technique. Therefore,
(19)$$\varepsilon =\frac{F_{53}}{1+a\lambda_{3}/R}$$
in III–V NWs is largely reduced with respect to the atomic V/III flux ratio in vapor $F_{53}$, particularly for thin NWs with $\lambda_{3}/R\gg 1$.

## 3. Results and Discussion

In our model, the effective V/III ratio is allowed to vary in the range $x+c_{g}\left(1-x\right)\leq \varepsilon \leq \infty$ to preserve the steady-state VLS growth conditions with a constant droplet volume. At $\varepsilon =x+c_{g}\left(1-x\right)$, the incoming group V and III fluxes equal each other, and all the arriving atoms are incorporated into the NW, meaning that the group V desorption fluxes are negligible. In this kinetic VLS regime, the vapor–solid distribution given by Equation (16) is reduced to the one-parametric Langmuir–McLean formula
(20)$$z=\frac{x}{x+c_{g}\left(1-x\right)}.$$

For a larger ε, a fraction of the A_2_ and B_2_ dimers must desorb from the droplet surface. In this case, the vapor–solid distribution is described by Equation (16), in which the liquid composition y should be calculated using Equation (3). The previously unknown $\chi_{A}+\chi_{B}$ in the parameter $\Gamma_{l}$ is now given by Equation (13). Therefore, $\Gamma_{l}$ becomes a function of y and x. Inferring the explicit dependence $y\left(x\right)$ from Equation (13) requires the solution of a quadratic equation for y. Substitution of the obtained $y\left(x\right)$ into Equation (16) yields the analytic vapor–solid distribution $z\left(x\right)$. This $z\left(x\right)$ is a function of vapor fluxes and the parameters of the liquid phase, which depend only on $\chi_{C}$. Therefore, the general vapor–solid distribution at intermediate ε contains a parametric dependence on $\chi_{C}$, which can be measured during [41] or after [9] growth.

This complicated procedure is not required for practical purposes. We now show that the parameters of liquid can be fully circumvented in the following approximation. The limiting behavior at ε→∞ corresponds to no-growth conditions where the arriving fluxes of A_2_ and B_2_ atoms are equalized by the desorption fluxes. In this case, the AC and BC pairs in liquid should also be at equilibrium with solid. The liquid–solid equilibrium in a ternary system corresponds to [13]
(21)$$\mu_{A}^{l}+\mu_{C}^{l}=\mu_{AC}^{s},\;\mu_{B}^{l}+\mu_{C}^{l}=\mu_{BC}^{s},$$
where $\mu_{AC}^{s}$ and $\mu_{BC}^{s}$ are the composition-dependent chemical potentials of the AC and BC pairs in solid. Using Equation (4) and the same expression for C atoms, $\mu_{C}^{l}=\mu_{C}^{l,0}+ln\chi_{C}+\psi_{C}$, along with the regular solution model for the chemical potentials in solid, $\mu_{AC}^{s}=\mu_{AC}^{s,0}+lnx+\omega {\left(1-x\right)}^{2}$ and $\mu_{BC}^{s}=\mu_{BC}^{s,0}+\ln\left(1-x\right)+\omega x^{2}$ (Refs. [10,11,12,13,14,15,16,17,18,19]), Equation (21) can be presented in the form
(22)$$\begin{array}{l} {\left(\chi_{A}\chi_{c}\right)}_{eq}=e^{-(\mu_{A}^{l,0}+\mu_{C}^{l,0}-\mu_{AC}^{s.0})-\psi_{A}-\psi_{C}}xe^{\omega {\left(1-x\right)}^{2}},\\ {\left(\chi_{B}\chi_{c}\right)}_{eq}=e^{-(\mu_{B}^{l,0}+\mu_{C}^{l,0}-\mu_{BC}^{s.0})-\psi_{B}-\psi_{C}}\left(1-x\right)e^{\omega x^{2}}. \end{array}$$

Upon substitution of these expressions into Equation (3), the simple calculation shows that the kinetic liquid–solid distribution is reduced to the equilibrium one [10,13,14,15]:(23)$$y_{eq}=\frac{1}{1+\beta_{l}\left(1-x\right)e^{\omega \left(2x-1\right)}/x},$$
where $\beta_{l}$ is the same as in Equation (3). For the equilibrium liquid–solid distribution, we have
(24)$$\frac{1}{1+\zeta {\left[\left(1-y_{eq}\right)/y_{eq}\right]}^{2}}=\frac{1}{1+f^{2}\left(x\right)},$$
where the equilibrium function $f\left(x\right)$ is the same as in Equation (5), and the affinity parameter is given by
(25)$$\beta_{g}=\sqrt{\frac{I_{B_{2}}^{0}}{I_{A_{2}}^{0}}}e^{\Delta \mu_{AC}^{0}-\Delta \mu_{BC}^{0}}.$$

Using the approximation $y=y_{eq}$ in Equation (16), the analytic vapor–solid distribution is obtained in the following form:(26)$$\begin{array}{l} z≅\frac{x}{\varepsilon }+\left[1-\frac{x+c_{g}\left(1-x\right)}{\varepsilon }\right]\frac{1}{1+f^{2}\left(x\right)},\\ f\left(x\right)=\beta_{g}\frac{\left(1-x\right)}{x}e^{\omega \left(2x-1\right)}, \end{array}$$
where $c_{g}$ is given by Equation (14) and $\beta_{g}$ is given by Equation (25). At $c_{g}=1$, it is reduced to the result of Ref. [19] given by Equation (4). If we re-write Equation (9) as
(27)$$\mu_{A_{2}}^{g}=2\mu_{A}^{l,0}+ln\left(\frac{I_{A_{2}}}{I_{A_{2}}^{eq}}\right),\;\mu_{B_{2}}^{g}=2\mu_{B}^{l,0}+ln\left(\frac{I_{B_{2}}}{I_{B_{2}}^{eq}}\right),$$
with $I_{A_{2}}^{eq}=\sigma_{A}I_{A_{2}}^{0}$ and $I_{B_{2}}^{eq}=\sigma_{B}I_{A_{2}}^{0}$ as the equilibrium fluxes at $\sigma_{A}=\sigma_{B}=1$, all the results remain, with $\beta_{g}$ modified to
(28)$$\beta_{g}=\sqrt{c_{g}}\sqrt{\frac{I_{B_{2}}^{eq}}{I_{A_{2}}^{eq}}}e^{\Delta \mu_{AC}^{0}-\Delta \mu_{BC}^{0}}.$$

Thus, the analytic vapor–solid distribution of VLS III–V ternary NWs based on group V intermix is given by Equation (26) and is very close to the vapor–solid distribution for III–V_x_-V_1−x_ materials grown in the vapor–solid mode without any droplet [19]. The main difference is in the ε parameter, which equals the atomic V/III flux ratio in vapor for the vapor–solid growth, while for VLS NWs it accounts for the fact that a catalyst droplet is able to collect many more group III atoms from the surrounding surfaces (as given, for example, by Equation (19)). The other difference is in the parameter $c_{g}$, which describes the effect of different condensation rates of A_2_ and B_2_ dimers into the droplet. These rates are usually assumed equal for the vapor–solid growth, corresponding to $c_{g}=1$. The obtained result is similar to Ref. [42], where it was shown that the vapor–solid distribution of VLS III–V ternary NWs based on group III intermix is kinetic, despite the fact that the corresponding liquid–solid distribution is close to equilibrium [10]. Equation (26) is approximate, because it uses the equilibrium shape of the liquid–solid distribution at intermediate ε which, strictly speaking, is valid only under no-growth conditions at ε→∞. A similar approximation was used in Ref. [19] for obtaining Equation (3).

The shape of the vapor–solid distribution given by Equation (26) is determined by the two thermodynamic parameters ω and $\beta_{g}$ and the two kinetic parameters $c_{g}$ and ε. The effective V/III ratio can easily be changed in the VLS growth experiments. The other parameters are independent of ε in the first approximation and determined primarily by the material system, growth catalyst and temperature. Figure 1 shows the vapor–solid distributions obtained from Equation (26) for a model system with a fixed ω= 1.6, $\beta_{g}=$ 0.3, $c_{g}=$ 2 and different ε. Although the miscibility gap is absent (ω<2), the equilibrium distribution and the distribution at ε=20 are non-linear. They are shifted to the right due to a small $\beta_{g}=0.3$, meaning that obtaining a noticeable fraction of the AC pairs in a NW requires a much larger fraction of the A atoms in vapor. As the effective V/III ratio decreases, the curves become closer to the kinetic Langmuir–McLean shape, which favors the vapor–liquid incorporation of the A atoms with respect to the B atoms at $c_{g}=$ 2. In principle, any vapor–solid distribution between the equilibrium and kinetic curves is possible and can be achieved by tuning the total V/III ratio at a fixed temperature (for example, by changing the total group V flux at a fixed group III flux). Regardless of the particular parameters used in Figure 1, the kinetically limited composition at small ε~1 and the thermodynamically limited composition at large ε≫1 must have different shapes, because they are controlled by the principally different physical parameters (describing either kinetic or equilibrium factors in the vapor–solid distribution). Increasing ε leads to excessive fluxes of group V atoms entering the droplet and leads to a transformation from a kinetic to an equilibrium shape of the distribution, with very different dependences of the NW composition on the vapor fluxes of the A and B atoms, as illustrated in Figure 1.

Such a behavior was observed in InSb_x_As_1−x_ epi-layers [2], AlSb_x_As_1−x_ epi-layers [20] and, more recently, in Au-catalyzed VLS InSb_x_As_1−x_ NWs [21]. These NWs were grown via MOVPE on InAs(111)B substrates at 450 °C using TMIn, TMSb and AsH_3_ precursors, with 50 nm diameter colloidal Au nanoparticles used as the VLS growth seeds. The total V/III flux ratio in vapor $F_{53}$ was set to 15, 27 and 56 by varying group V fluxes at a constant TMIn flux. These vapor–solid distributions were analyzed in our recent work [19]. Here, we extend the analysis by considering the vapor–solid distributions of InSb_x_As_1−x_ NWs together with epi-layers that were grown concomitantly with the NWs [21]. Figure 2 shows the measured vapor–solid distributions of InSb_x_As_1−x_ NWs and epi-layers. The ω value at 450 °C is well known and equals 1.566 [19,43]. The vapor–solid growth of epi-layers at a high $F_{53}$ of 27 must yield a close-to-equilibrium shape of the corresponding distribution. This allows us to choose a $\beta_{g}$ value of 0.34, which is close to the equilibrium constant of 0.429 given in Ref. [1] and used for modeling in Ref. [21]. The kinetic curve, obtained for NWs at $F_{53}=15$, is linear. This should correspond to $c_{g}=1$, that is, equal incorporation rates of Sb and As into the droplet. Assuming that $\beta_{g}$ is the same for epi-layers and NWs (which is not guaranteed in the general case), the different behaviors of the vapor–solid distributions in Figure 2 are entirely due to the different ε values in Equation (26). For epi-layers, the fitting value of ε=27 is the same as $F_{53}$ in vapor. For NWs, the fitting values of ε are 11–16 times smaller than $F_{53}$ in vapor, which is explained by the additional fluxes of diffusive In adatoms from the surrounding surfaces as compared to the surrounding vapor. This observation was made in the original work [21].

Before discussing the data on VLS InP_x_As_1−x_ and GaP_x_As_1−x_ NWs, we note that the parameter $\beta_{g}$ given by Equation (25) or Equation (28) contains the exponential of the well-known difference of chemical potentials for pure binaries $\Delta \mu_{AC}^{0}-\Delta \mu_{BC}^{0}$ [44,45,46], while the pre-exponential factor (for example, ${\left[c_{g}I_{B_{2}}^{eq}/I_{A_{2}}^{eq}\right]}^{1/2}$ in Equation (28)) is less obvious. It is different from what is usually considered in the equilibrium constants for surface reactions [1,2,20]. These constants describe the equilibrium of binary or more complex vapors with binary solids, while our $I_{A_{2}}^{eq}$ and $I_{B_{2}}^{eq}$ are the equilibrium fluxes for pure group V liquids. Our $\beta_{g}$ also includes the unknown parameter $c_{g}$. An accurate analysis of these factors is beyond the scope of this work. In what follows, we will use $\beta_{g}$ as a fitting value but take into account the thermodynamic trend that follows from the exponential factor $exp\left(\Delta \mu_{AC}^{0}-\Delta \mu_{BC}^{0}\right)$ in the affinity parameter.

Figure 3 shows the vapor–solid distributions of Au-catalyzed InP_x_As_1−x_ NWs obtained by Persson and coauthors [26]. These NWs were grown via CBE on InAs(111)B substrates using 50 nm diameter colloidal Au droplets, which resulted in ~60 nm diameter NWs. The growth started with InAs NW stems and continued with InP_x_As_1−x_ sections grown at three different temperatures of 390 °C, 405 °C and 435 °C. The total V/III flux ratio in vapor during the growth of InPAs sections was in the range from 30 to 45. It is seen that the values of z are systematically larger than x, meaning that the incorporation of P atoms is lower than that of As atoms. The authors fitted the data using the kinetic Langmuir–McLean Equation (20) with the low $c_{g}$ values that increased from 0.105 at 390 °C to 0.175 at 435 °C (dashed lines in Figure 3). The values of $exp\left(\Delta \mu_{InP}^{0}-\Delta \mu_{InAs}^{0}\right)$ equal 0.233 at 390 °C, 0.2375 at 405 °C and 0.244 at 435 °C [44,45,46]. This shows a thermodynamic trend for having a smaller fraction of P atoms in vapor than in solid in the whole temperature domain studied in Ref. [26]. The very high V/III flux ratios employed in this work should lead to desorption of the excessive P and As atoms from the droplet surface, as in the previous case of InSb_x_As_1−x_ NWs. Therefore, we fit the data using the general equation (26), using ε values that are noticeably larger than unity. They appear close to InSbAs NWs under similar V/III flux ratios in vapor. The best fits are obtained with $\beta_{g}=0.1$ at 390 °C, 0.13 at 405 °C and 0.2 at 435 °C, and $c_{g}=1$ in all cases (solid lines in Figure 3). These curves provide slightly better fits than the Langmuir–McLean formula.

It is interesting to note that these fitting values are very close to the effective ratios of the P-over-As incorporation rates obtained in Ref. [26]. This is most probably explained by the relatively weak interactions of InP and InAs pairs in solid, corresponding to the low ω values given in Table 1. In this case, the equilibrium distribution in Equation (26) is close to the Langmuir–McLean shape. This property follows directly from Equation (26) for $f\left(x\right)$ at ω→0. Therefore, fitting the vapor–solid distributions of III–V ternary NWs with low pseudo-binary interaction parameters ω by the one-parametric Langmuir–McLean formula is entirely possible [13,14,15,47]. The effective ratio of the incorporation rates of different group V atoms must, however, include the differences in the desorption rates and the dependence on the total V/III flux ratio, as in our model.

GaPAs is another example of a ternary material with low ω, which are in the range from 0.64 to 0.7 in the typical growth temperature window of 550–630 °C (see Table 1). In contrast to InPAs, the difference of the chemical potentials $\Delta \mu_{GaP}^{0}-\Delta \mu_{GaAs}^{0}$ is positive, yielding values of $exp\left(\Delta \mu_{GaP}^{0}-\Delta \mu_{GaAs}^{0}\right)$ ranging from 1.849 at 550 °C to 1.782 at 630 °C [44,45,46]. This should favor faster incorporation of P atoms relative to As atoms and, consequently, a larger P fraction in vapor relative to solid in close-to-equilibrium growth regimes under high V/III flux ratios. Figure 4a,b show the compilation of the vapor–solid distributions of VLS GaP_x_As_1−x_ NWs from the four works. Metaferia and coauthors grew the NWs via Au-catalyzed MOVPE using the substrate-free aerotaxy at 550 °C, under low total V/III flux ratios in vapor from 0.82 to 1.64 [32]. Other GaP_x_As_1−x_ NWs [28,30] or GaP_x_As_1−x_ sections in GaP NWs [31] were grown via the self-catalyzed MBE (with Ga droplets) on Si(111) substrates. Himwas and coauthors [28] grew the NWs at 610 °C under total V/III flux ratios ranging from 10 to 12. Zhang and coauthors [30] and Bolshakov and coauthors [31] grew the NWs at 630 °C under higher total V/III ratios, ranging from 40 to 80 in Ref. [30] and from 16 to 32 in Ref. [31]. Different procedures for preparation of the Ga droplets were used and resulted in different NW surface densities, diameters and lengths. The vapor–solid distribution of the NWs grown by aerotaxy at low V/III ratios corresponds to a lower incorporation rate of the P atoms, while the other NWs grown at much higher V/III ratios exhibit the opposite trend. The vapor–solid distributions obtained by Zhang and coauthors [30] and Bolshakov and coauthors [31] at 630 °C are very close to each other.

Figure 4a shows the fits to the whole set of data obtained from Equation (26) using different ε. The data of Ref. [32] at low $F_{53}$ are fitted with a minimum ε corresponding to the Langmuir–McLean shape at $c_{g}=0.27$, as in the original work. The MBE data of Refs. [28,30,31] are fitted with large values for ε of 1.8, 2.97 and 4.5, using the same parameter $\beta_{g}=3.1$ in the equilibrium distribution and the same $c_{g}=0.27$. The value of $c_{g}$ is not critical for these fits. The MBE data can be well fitted using, for example, $c_{g}=1$ having slightly different ε values. This figure shows the same trend as in Figure 1 and Figure 2, that is, transitioning of the kinetic distribution to the equilibrium shape when the total V/III ratio is increased. The purely kinetic black curve at ε~1 is transformed to more thermodynamically limited curves at larger ε. The difference between the three curves at 610 °C and 630 °C is not due to slightly different growth temperatures but rather to different effective V/III flux ratios entering the droplet. It is noteworthy that the trends shown in Figure 2 for InSb_x_As_1−x_ NWs and in Figure 4a for GaP_x_As_1−x_ NWs are different. In both cases, the shapes of the vapor–solid distributions are transitioned from the kinetic to the thermodynamically limited for larger V/III flux ratios. However, in the InSb_x_As_1−x_ system, the $x\left(z\right)$ curve shifts to the right and become non-linear when the V/III flux ratio is large, meaning that thermodynamic factors lead to the suppression of the Sb incorporation (see the dashed equilibrium curve in Figure 2). In the GaP_x_As_1−x_ system, the situation is reversed, with the $x\left(z\right)$ dependences shifting to the left for larger ε. In this case, the incorporation of the P atoms is favored by thermodynamics, as described by the equilibrium vapor–solid distribution shown by the dashed curve in Figure 4a.

Figure 4b shows that equally good fits can be obtained using the Langmuir–McLean formula with different $c_{g}$ for all the data. The fitting value of the effective ratio of the incorporation rates of P over As atoms increases from 0.27 to 4.05 (the fitting value of 2.97 was obtained by Zhang and coauthors in Ref. [30] for their data). It would be difficult to explain this trend without considering desorption of the excessive group V atoms in the MBE growths under very high V/III ratios. As in the previous case, the Langmuir–McLean shapes provide excellent fits due to the low ω values in this material system.

## 4. Conclusions

In summary, we have shown that the uncertainty in the unmeasurable group V concentrations in a catalyst droplet can be fully circumvented by considering the growth kinetics of VLS III–V ternary NWs based on group V intermix in the entire VLS growth process rather than in the liquid–solid growth alone. The self-consistent vapor–solid distribution of VLS III–V NWs is close to the shape obtained earlier for the vapor–solid growth without any droplet. The simple analytic form of the distribution is useful for the analysis of the data, and it fits well the available data on InSb_x_As_1−x_, InP_x_As_1−x_ and GaP_x_As_1−x_ NWs grown via different epitaxy techniques at different temperatures and having different metal catalysts.

This work presents the first attempt to obtain a general vapor–solid distribution of VLS III–V ternary NWs based on group V intermix beyond the common one-parametric Langmuir–McLean approach. This enables us to model and explain some compositional trends which are inaccessible in the common approach, in particular, the strong dependence of the vapor–solid distribution on the total V/III flux ratio. We have used several approximations in deriving the final result. The most important approximation is the effective absence of desorption or downward diffusion of group III atoms from the droplet. This process may become important at higher temperatures. If such a sink of group III atoms is present, the governing equations should include the outgoing flux of group III atoms, which depends on the group III concentration in the droplet. The size of droplets at the NW tip, which is known to affect the composition of VLS III–V ternary NWs [10,29], is described in our model by the time-independent NW radius R and the droplet contact angle β. More complex geometries, such as tapered NWs, have not been studied. We plan to consider these refinements in a forthcoming work. It will be interesting to test the model against the data on VLS NWs of ternary III–V materials with the miscibility gaps at a growth temperature, for example, GaSb_x_As_1−x_ or AlSb_x_As_1−x_ NWs, where the miscibility gaps can be suppressed by tuning the V/III flux ratio. Overall, this simple model for the complex ternary material system should be useful for understanding and tuning the compositions of different III–V NWs based on group V intermix, and it may be extended to other material systems using highly volatile growth species.

## Figures and Tables

**Figure 1 nanomaterials-14-00207-f001:**
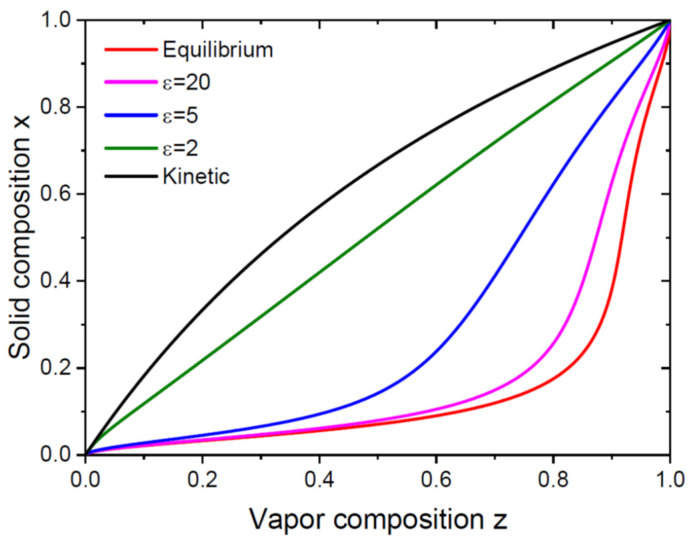
Transformation of the vapor–solid distribution from the equilibrium shape at ε→∞ to the purely kinetic Langmuir–McLean shape at $\varepsilon =x+c_{g}\left(1-x\right)$. Any distribution between the equilibrium and kinetic curves is possible and is regulated by the single parameter ε.

**Figure 2 nanomaterials-14-00207-f002:**
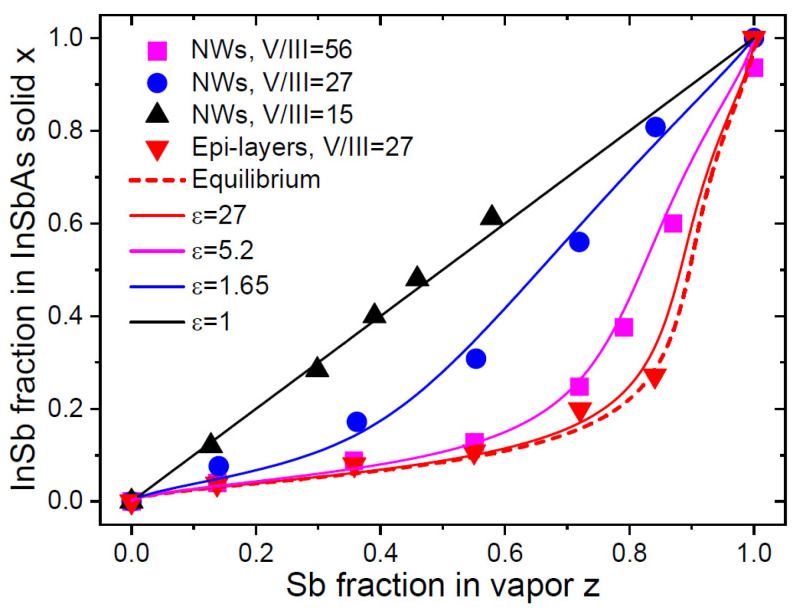
Vapor–solid distributions of Au-catalyzed VLS InSb_x_As_1−x_ NWs and epi-layers grown via MOVPE at 450 °C [21] (symbols), fitted by Equation (26) using the parameters given in Table 1 (solid lines). Dashed line shows the equilibrium distribution.

**Figure 3 nanomaterials-14-00207-f003:**
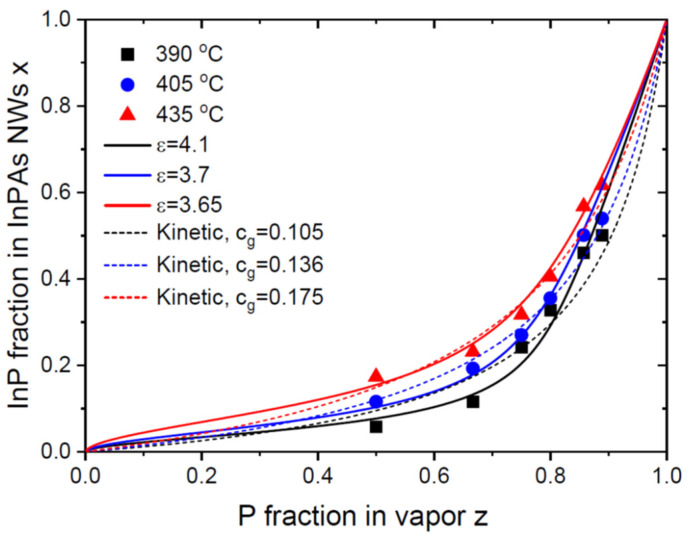
Vapor–solid distributions of Au-catalyzed VLS InP_x_As_1−x_ NWs grown via CBE on InAs NW stems at 390 °C, 405 °C and 435 °C [21] (symbols). Solid lines are the fits obtained from Equation (26) using the parameters listed in Table 1. Dashed lines show the fits obtained from Equation (20) using different parameters, $c_{g}$, given in the legend.

**Figure 4 nanomaterials-14-00207-f004:**
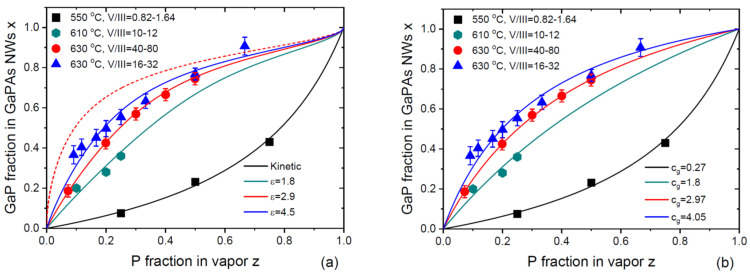
(**a**) Vapor–solid distributions of VLS GaP_x_As_1−x_ NWs grown via the substrate-free Au-catalyzed aerotaxy at 550 °C at low V/III flux ratios in vapor $F_{53\;}\sim 1$ [32], Ga-catalyzed MBE on Si(111) substrates at 610 °C at $F_{53\;}=$ 10–12 [28], 630 °C at $F_{53\;}=$ 40–80 [30], and 630 °C at $F_{53\;}=$ 16–32 [31]. Solid lines are the fits by Equation (26) using the parameters given in Table 1. Dashed line shows the equilibrium distribution at 630 °C, which is almost indistinguishable from the one at 610 °C. The data of Ref. [32] are fitted by the Langmuir–Mclean Equation (20) with $c_{g}=$ 0.27. (**b**) Same data as in (**a**), fitted by Equation (20) using different $c_{g}$ given in the legend.

**Table 1 nanomaterials-14-00207-t001:** Parameters of III–V ternary epi-layers and VLS NWs based on group V intermix.

Material	Catalyst	T (°C)	V/III Ratio in Vapor $F_{53}$	ε	ω	$\beta_{g}$	$c_{g}$
InSb_x_As_1−x_ layers [21]	-	450	27	27	1.566	0.34	1
InSb_x_As_1−x_ NWs [21]	Au	450	56	5.2	1.56	0.34	1
			27	1.65			
			15	1			
InP_x_As_1−x_ NWs [26]	Au	390	30–45	4.1	0.546	0.1	1
		405		3.7	0.534	0.13	1
		430		3.65	0.515	0.2	1
GaP_x_As_1−x_ NWs [32]	Au	550	0.82–1.64	min	0.703	-	0.27
GaP_x_As_1−x_ NWs [28]	Ga	610	10–12	1.8	0.656	3.1	0.27
GaP_x_As_1−x_ NWs [30]	Ga	630	40–80	2.9	0.641	3.1	0.27
GaP_x_As_1−x_ NWs [31]	Ga	630	16–32	4.5	0.641	3.1	0.27

## Data Availability

Data are contained within the article.

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
