# Peer review of "Circumventing the Uncertainties of the Liquid Phase in the Compositional Control of VLS III–V Ternary Nanowires Based on Group V Intermix"

_nanomaterials, 2024, doi:10.3390/nano14020207_

Round 1

Reviewer 1 Report

Comments and Suggestions for Authors

This manuscript reported a self-consistent growth model of III-V ternary nanowires, which circumvents the uncertainties in the infinitely low group V concentrations in the droplet. This work is interesting and of great significance for the composition controlled synthesis of III-V nanowires. Nevertheless, some modifications are suggested to make this manuscript more easily understandable. After the following explanations and alterations are handled, I would be pleased to recommend its acceptance to publish in this journal.

1.    The logic of the introduction part is not smooth, it is suggested that the application prospect of nanowires is proposed first, and then the importance of composition control is emphasized, and then the research content of this manuscript is introduced.

2.    Figure 1 shows a clear difference between the green line (𝜀=2) and the blue line (𝜀=5), and the author should add what the reason for this transformation is in the manuscript, as this is the critical point of the transformation.

3.    In Figure 4 (a), the dashed line representing equilibrium distribution at 610 °C is not shown. Moreover, the manuscript describes “which is almost indistinguishable from the one at 610 °C”, why are the solid lines so different (red line, blue line and green line) and the dashed lines almost indistinct?

Author Response

The response letter is attached.

Reviewer 2 Report

Comments and Suggestions for Authors

The manuscript focuses on the understanding of the growth of ternary III-V nanowires with group V intermix, which have attracted a lot of research interest lately. One important reason is that the band structure of the ternary nanowires can be continuously tuned by composition engineering. But the incorporation of group V elements during vapor-liquid-solid (VLS) growth is a complicated and still not well-understood process. The composition of the final nanowires depends on a range of growth parameters. The work presented in the manuscript deals with this challenge and establishes a theoretical framework that directly relates vapor phase composition of the source gases, which can be relatively easily monitored experimentally, with material composition in the grown solid nanowires, thus circumventing the uncertainties of liquid phase in the compositional control of VLS III-V ternary nanowires based on group V intermix. The topic of the manuscript is important and timely. The method used is scientifically sound. The results are well presented, and the manuscript is well written. I recommend the publication of the manuscript after some comments are addressed.

Here are more detailed comments:

1.     The trend shown in Figure 2 is different from that shown in Figure 4(a). It is stated throughout the manuscript that, at high V/III ratio, x(z) is close to the equilibrium case. In this case, x(z) curve is nonlinear and is located to the right of the plot, as shown in Figure 2. As V/III decreases, the x(z) curve shifts to the left and even becomes linear when V/III is small. However, in Figure 4(a), the trend for x(z) as a function of V/III ratio is not the same. For example, the curve corresponds to the smallest V/III is the one to the right in the plot and nonlinear. The curves for other V/III ratios also don’t follow the same trend as shown in Figure 1 and 2. Of course, the data shown in Figure 4 are acquired at slightly different temperatures. But please explain why the trends shown in Figure 2 and 4 are different and still can be captured by the current model.

2.     The model established in the present work apparently does not consider some parameters related to the liquid droplets at the tip of the nanowires, such as the size and shape of the droplets. These parameters have been shown to affect the composition of III-V nanowires grown by VLS process (reference: Nano Lett. 2016, 16, 1237–1243). Please discuss the effect of these parameters or justify the omission of them.

Author Response

The response letter is attached.

Round 2

Reviewer 1 Report

Comments and Suggestions for Authors

There is no problem.